# Regionality of short and long period oscillators in the suprachiasmatic nucleus and their manner of synchronization

**Tadamitsu Morimoto[1], Tomoko Yoshikawa[2], Mamoru Nagano[1], Yasufumi Shigeyoshi[1]***

**1** Department of Anatomy and Neurobiology, Graduate School of Medicine, Kindai University, Osaka-Sayama, Osaka, Japan, **2** Organization for International Education and Exchange, University of Toyama, Toyama, Japan

*shigey@med.kindai.ac.jp

**Data Availability Statement:** Data used to generate the figures and supplementary figures in this manuscript are accessible on a public figshare repository (DOI: 10.6084/m9.figshare.21261324).

## Abstract

In mammals, the center of the circadian clock is located in the suprachiasmatic nucleus (SCN) of the hypothalamus. Many studies have suggested that there are multiple regions generating different circadian periods within the SCN, but the exact localization of the regions has not been elucidated. In this study, using a transgenic rat carrying a destabilized luciferase reporter gene driven by a regulatory element of *Per2* gene (*Per2*::dLuc), we investigated the regional variation of period lengths in horizontal slices of the SCN. We revealed a distinct caudal medial region (short period region, SPR) and a rostro-lateral region (long period region, LPR) that generate circadian rhythms with periods shorter than and longer than 24 hours, respectively. We also found that the core region of the SCN marked by dense VIP (vasoactive intestinal peptide) mRNA-expressing neurons covered a part of LPR, and that the shell region of the SCN contains both SPR and the rest of the LPR. Furthermore, we observed how synchronization is achieved between regions generating distinct circadian periods in the SCN. We found that the longer circadian rhythm of the rostral region appears to entrain the circadian rhythm in the caudal region. Our findings clarify the localization of regionality of circadian periods and the mechanism by which the integrated circadian rhythm is formed in the SCN.

## Introduction

The center of the mammalian circadian clock is located in the suprachiasmatic nucleus (SCN) of the hypothalamus, which consists of a bilateral pair of SCN across the third ventricle and contains approximately 10,000 cells on each side [1]. A mutual positive/negative feedback loop is formed by the regular expression of multiple clock genes: *Per1*, *Per2*, *Cry1*, *Cry2*, *Bmal1*, *Clock*, and their protein products [1, 2]. The clock gene expression rhythm within the SCN is robust, having a circadian rhythm of approximately 24 hours [2]. To maintain the integrated circadian rhythm as a single functional unit, the circadian rhythms in the oscillating neurons in the SCN must be synchronized.

Other data and further information are also available from the corresponding author on reasonable request.

**Funding:** The research was supported in part by JSPS KAKENHI (Grant-in-Aid for Scientific Research(C), Grant No; 19K06774 to T.Y., 20K07234 to N.M., 17K08580 to Y.S.). The funders had no role in study design, data collection and analysis, decision to publish, or preparation of the manuscript.

**Competing interests:** The authors have declared that no competing interests exist.

Functionally the SCN is divided into two regions [3]. The ventrolateral core region receives direct projection from the retina, while the dorsomedial shell region does not [3, 4]. The core region is composed of photo-responsive retinorecipient neurons that deliver photic information to the shell [3, 5]. After an abrupt shift of the light/dark cycle (LD cycle), the locomotor activity of rodents shows a slow shift in locomotor activity that is observed as jet lag [6]. We previously found a slow shift of the circadian rhythm in the shell after an abrupt shift of the LD cycle, and supposed that the slow shift causes jet lag [3].

The SCN is a heterogeneous structure comprising many types of neurons [1, 4, 7, 8]. Most SCN neurons are GABAergic, which is an inhibitory neurotransmitter [1, 4, 9]. Many of these GABAergic neurons co-express neuropeptides such as vasoactive intestinal peptide (VIP), gastrin releasing polypeptide (GRP), and arginine vasopressin AVP) [1, 10]. AVP is expressed primarily in the shell of the SCN, while VIP and GRP are expressed in the core. VIP has been demonstrated to be particularly important for the maintenance and entrainment of cellular clocks in individual SCN neurons [11–14]. In addition, AVP-expressing neurons are densely expressed in the shell and have been shown to extend jet lag [15, 16]. Other neurotransmitters such as GABA and GRP may play additional roles for the maintenance of the circadian rhythm in the SCN [17–20].

Previous studies have suggested that each neuron in the SCN has a different cell-autonomous circadian rhythm [21, 22], and that there are regional period differences [21, 23–26]. Noguchi et al. [25, 26] dissected the SCN into dorsal-ventral and rostral-caudal coordination, and found differences in circadian period within the SCN. Koinuma et al. [21] revealed that there is a small region in the dorsomedial part of the ex-vivo coronal slices of the SCN, showing a shorter circadian period (short period region, SPR) than the rest of the SCN (long period region, LPR) and also revealed that a phase wave propagates from SPR to LPR. However, the localizations of these regions generating the various circadian rhythms within the SCN and how they are synchronized with each other has not been fully elucidated.

In this study, we investigated differences of circadian period in the rat SCN by monitoring the bioluminescence of coronal and horizontal slices. We observed and analyzed the regional period differences in the SCN, rostral-caudal coordination, and the relationship between the direction of the phase wave propagation and the period regionality. Furthermore, by dissecting the SCN slice into fragments, we investigated which region of the SCN determines the circadian period of the whole SCN and how the circadian rhythms are integrated within the SCN.

## Materials and methods

### Animals

Male transgenic rats of Wistar background carrying a bioluminescence reporter of *Period2* (*Per2*) expression were used. In these rats, the rat *Per2* promoter region was fused to a destabilized luciferase (*dLuc*) reporter gene [27]. The rats were bred and raised in our animal facility in Kindai University Faculty of Medicine under LD cycle with lights on/off at 7:00/19:00 or 19:00/7:00. Light intensity during the light phase was approximately 400 lux. Room temperature was 22 ˚C. The rats were fed commercial chow and tap water ad libitum. All the rats were two to three months old at the time of the experiments. All procedure was performed under isoflurane anesthesia, and all efforts were made to minimize suffering.

The experiments were conducted in accordance with the Kindai University Animal Experiment Regulations and the NIH Guidelines for the Care and Use of Laboratory Animals. All animal experimental procedures were approved by the Institutional Animal Experimentation Committee of Kindai University School of Medicine (Permission number: KAME-30-036).

## Slice preparation for ex-vivo cultures

Under deep anesthesia, an animal was decapitated between ZT0 and ZT12 (ZT, zeitgeber time), and the brain was harvested in ice-cold Hanks' balanced salt solution (pH 7.4, Sigma, USA). Coronal and horizontal brain slices were prepared by a Microslicer (Dosaka, Japan) at thicknesses of 200 μm and 150 μm, respectively. The region containing the SCN was dissected from the slices and placed on a culture insert (ORG50; Millipore, Germany) in 35-mm culture dishes with 1.3 mL of culture medium, DMEM (12100046, Gibco, USA) containing D-luciferin K salt (0.1 mM for PMT recording, 0.2 mM for imaging; DOJINDO, Japan) and supplemented with $NaHCO_3$ (2.7 mM; Nacalai tesque, Japan), HEPES (10 mM; DOJINDO, Japan), kanamycin (20 mg/L; Gibco, Thermo Fisher Scientific, USA), insulin (5 μg/mL; Sigma, USA), putrescine (100 nM; Sigma, USA), apo-transferrin (100 mg/mL; Sigma, USA), progesterone (20 nM; Sigma, USA), and sodium selenite (30 nM; Gibco, USA) [28, 29].

## In situ hybridization

Digoxigenin-labeled r*Vip* (nucleotides 119–808; accession number X02341) cRNA probes were synthesized according to the manufacturer's protocol (Roche Diagnostics Japan, Japan). Horizontal brain slices (150 μm) were prepared as described above, fixed by immersing in 4% paraformaldehyde solution overnight, and processed using the free floating in situ hybridization method as described in our previous studies [6, 30]. In the present study, using this in situ hybridization technique, we detected the core region by the localization of *Vip* mRNA-containing neurons in horizontal SCN slices.

## Bioluminescence recording of coronal slices by PMT

Bioluminescence from cultured coronal slices of 200 μm thickness was measured using a photomultiplier tube (PMT, Kronos; ATTO, Japan) for 1 in every 10 min at 37 ˚C. The measurements were started immediately after the slice preparation and continued for 7–14 days. The data between 12 and 132 hours in culture were used for analysis.

## Bioluminescence recording of horizontal slices by EMCCD camera

Horizontal slices of 150 μm thickness were cultured under the same conditions except that the concentration of D-luciferin K salt was 0.2 mM. *Per2*::dLuc luminescence was recorded by one of three imaging systems: Multiversa, (ATTO, Tokyo, Japan) with an EMCCD camera (iXon 897, Andor, Belfast, UK; Exposure: 59 min., Em gain value: 500, Binning: 1×1) cooled at -90 ˚C; LUMINOVIEW (LV200, OLYMPUS, Japan) with an EMCCD camera (C9100-23B, Hamamatsu Photonics, Japan; Exposure: 29–59 min., Sensitivity gain: 150–200 (exposure 29 min.), 100–150 (exposure 59 min.), Gain: 1) cooled at -80 ˚C; or Cellgraph with an EMCCD camera (AB-3000, Atto, Japan; Exposure: 59 min., Electron Multiplier Gain: 300, Pre-Amplifier Gain: 1.0) cooled at -70 ˚C. Bioluminescence was recorded every 30 or 60 min. The measurement was started immediately after slice preparation and continued for 7 to 10 days. After the measurement, an integrated image of 24 to 120 hours was created using Image J, and the outline of the SCN was obtained from this image.

## Analysis of bioluminescence data to reveal the period and phase of the circadian rhythm

We set ROIs (ROI, regions of interest) dividing the SCN horizontal slices into several regions depending on the experiment. The average value of bioluminescence inside each ROIs was measured. The SCN horizontal slice was further divided into small square grids (grid size;

32 μm × 32 μm). The raw data from PMT and cooled CCD camera were detrended by subtracting the 24-hour moving average [31, 32] and smoothed by taking a 5-point moving average [33]. The detrended and smoothed data from 24 hours to 120 hours after the beginning of recording were fitted to a mathematically generated damped cosine curve [21] {$y = a + b•\exp(-c•t) •\cos 2\pi[(t + d) / e]$, $t$: time, $a$: mesor, $b$, c, $d$: constants, $e$: period} using Excel Solver (Microsoft, USA). We calculated the period and phase of the circadian bioluminescence rhythm from each grid via the fitted curve. Grids with correlations <0.6 between detrended bioluminescence data and fitted curve were excluded from the analysis. Origin (OriginLab, USA) was used to visualize the circadian period from each grid.

## Statistical analysis

Repeated measures one-way ANOVA with post-hoc Bonferroni test was used to analyze the period length measured by PMT in coronal slices. To analyze the effect of forskolin (FK, adenylate cyclase activator), we conducted repeated measures two-way ANOVA with post-hoc Bonferroni test, and multivariate comparison with post-hoc Tukey test. To analyze the effect of separation by a scalpel, we conducted repeated measures two-way ANOVA with post-hoc Bonferroni test.

## Results

### Phase mapping of circadian oscillations in horizontal slices of rat SCN

As shown in Fig 1A, horizontal SCN slices of 150 μm thickness were prepared (n = 4). In a slice, to examine how a phase-advanced or phase-delayed region relates to the core and the shell regions in the SCN, we investigated the localization of *Vip*-mRNA expressing neurons as a marker of the SCN core by using in situ hybridization [6] (Fig 1B). Then we examined the *Per2*::*dLuc* bioluminescence rhythm focusing on the difference along the rostro-caudal axis within the rat SCN, using a different individual from the one presented above. The phase wave of bioluminescence propagated from caudal to rostral and from medial to lateral in the SCN (Fig 1C, S1 Movie). The bioluminescence rhythm in the caudal area and in the medial area were the most advanced (Fig 1D and 1E, red) and those of the rostral and lateral areas were the most delayed (Fig 1D and 1E, blue). To examine this phase distribution within the SCN in more detail, we divided the SCN into small grids and analyzed them separately (Fig 1F). The phase was advanced in the medial-caudal area relative to the lateral-rostral area of the SCN. Comparing the phase map in the SCN horizontal slice (Fig 1F) with the localization of *Vip* mRNA expressing neurons (Fig 1B), *Vip*-expressing regions appeared to correspond to the regions with delayed phase within the SCN. In contrast, the shell included regions with both advanced and delayed phase.

### Circadian period analysis on consecutive coronal slices of SCN

We next investigated the difference in circadian period using consecutive coronal sections containing the SCN. Six consecutive 200 μm coronal slices were prepared and were set into Kronos for examination of the *Per2*::dLuc circadian bioluminescence rhythm (n = 6). Among them, clear circadian rhythms were detected from three or four SCN slices per animal. The circadian rhythm showing the largest amplitude was selected (Middle) along with the adjacent rostral (Rostral) and caudal sections (Caudal) (Fig 2A, S1 Fig). The mean values of the period length of the Rostral, Middle, and Caudal were 23.8 ± 0.2, 23.8 ± 0.1, and 22.4 ± 0.2 hours, respectively (Mean ± SE, Fig 2B). The circadian period of Caudal was significantly shorter than those of the other two sections (Fig 2B), while no significant difference was detected

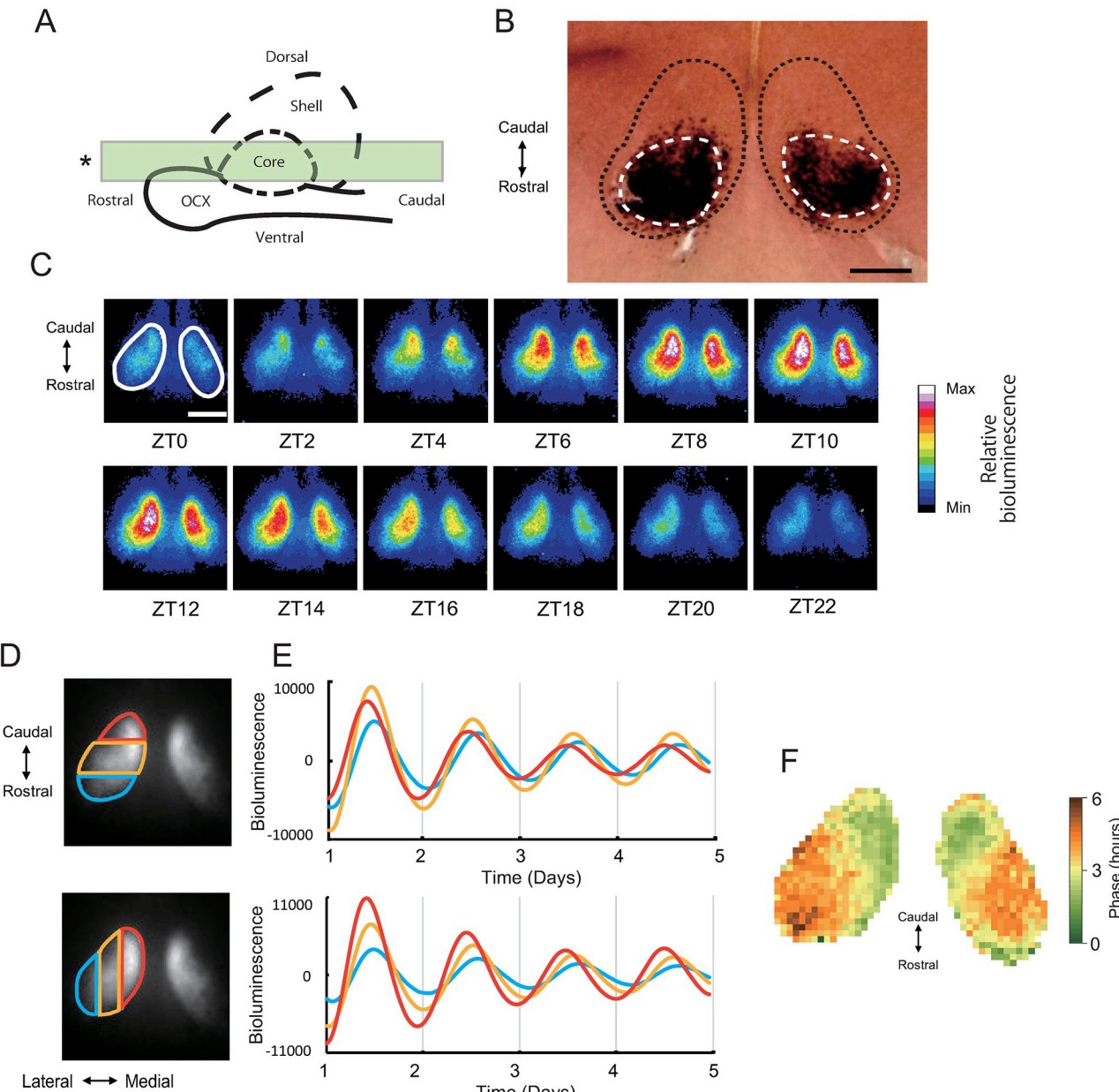

**Fig 1. Schema of the SCN slices and *Per2*::dLu*c* bioluminescence rhythm in the SCN.** (A) A schematic diagram of the sagittal section of the SCN (modified from Nagano et al., 2019). The outline of the SCN is denoted by a dashed line (considered as the shell region of the SCN), and the outline of the VIP region (considered as the core region of the SCN) by a dashed line. The green rectangle (*) in the picture indicates the location of the horizontal slice excision for the present study. (B) Representative horizontal section of the SCN showing the r*Vip*-expressing neurons by in situ hybridization. The outline of the core region was identified by r*Vip*-expressing neurons. Black dashed line: outline of the SCN, White dashed line: core region of the SCN, Scale bar: 250 μm. (C) Representative bioluminescence images of a horizontal slice of the rat SCN. White line indicates the outline of the SCN. The beginning of the first light period in the former light-dark cycle before decapitation was described as (projected) ZT0. Scale bar: 500 μm. Data shown in C-F are from a single slice. (D) three ROIs were set on a unilateral SCN with the same rostro-caudal width and same medial-lateral width. (E) The average of the bioluminescence intensity of each ROI was plotted against time. The phase in the caudal ROI (red) are advanced compared to those of the middle (yellow) and rostral (blue) fragments (upper panel). The phases in the medial ROIs (red) are advanced compared to those of the middle (yellow) and lateral (blue) fragments (lower panel). (F) Phase map of 1st acrophase (after the first pZT0) of the SCN. This figure indicates that the phase wave propagates from the green region to the brown region. The phase was advanced in caudal and medial relative to the rostral and lateral, indicating that the phase wave propagates from the caudal side to the rostral side, and from the medial side to the lateral side. Grid size: 32 μm.

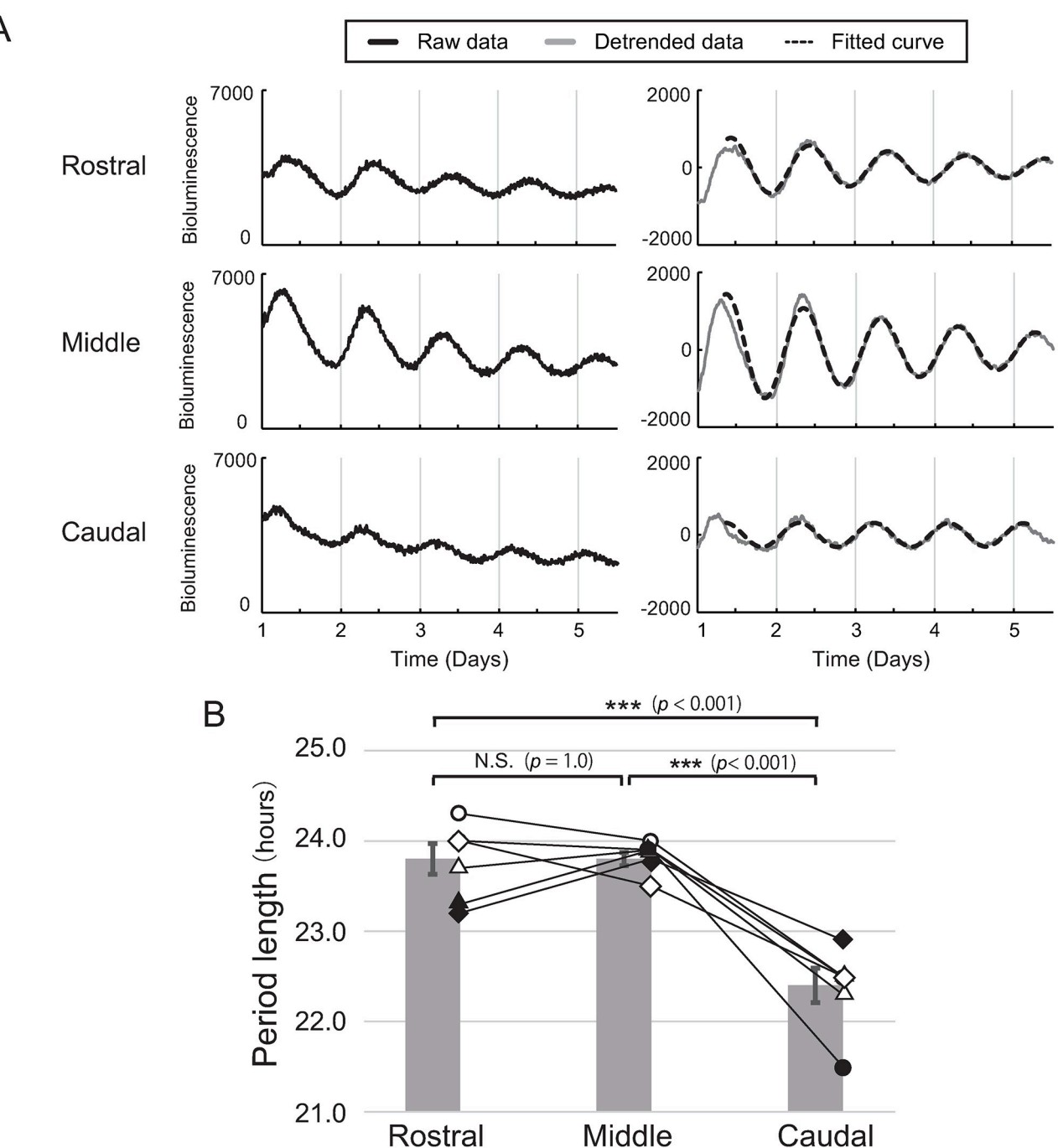

**Fig 2. *Per2*::dLuc bioluminescence rhythm in coronal slices of the SCN measured by PMT.** (A) Representative *Per2*::dLuc bioluminescence rhythm from consecutive slices of a single individual animal (Rostral, Middle, and Caudal). Left and right panels show raw data and detrended data, respectively. (B) Circadian periods from Rostral, Middle, and Caudal (Mean ± SE). The periods of each slice are superimposed. Period data from the same individuals are connected by lines. Repeated measures one-way ANOVA with Post-hoc Bonferroni test. ***: $p < 0.001$ vs Rostral and Middle, N. S.: not significant.

between Rostral and Middle ($p$ = 1.0) (one-way ANOVA, F(2,4) = 23, p = 0.0018; post-hoc Bonferroni test, caudal vs rostral, p = 0.0055; caudal vs middle, p = 0.0039; rostral vs middle, p = 1.0). These findings suggest that the caudal region of the SCN contains oscillators with shorter circadian periods than those in the middle and rostral regions.

## Effect of forskolin on circadian period in horizontal slice

To confirm the regional differences of circadian period, we disrupted the intracellular synchronization using *forskolin* (FK). Previously we found that FK disrupts the intercellular synchronization in the SCN [21, 34]. Horizontal slices of 150 μm thickness were cultured in medium containing 10 μM FK (n = 6) or vehicle (DMSO). One to three horizontal slices were placed on a single culture insert, and bioluminescence was recorded by one of the EMCCD cameras. We set regions of interest (ROI) on the rostral half and caudal half of the unilateral SCN (Fig 3A). and designated them as Rostral area and Caudal area, respectively (Fig 3A). In FK treated horizontal slices, we found the circadian period of the Caudal area to be significantly shorter than that of the Rostral area (Fig 3B, S2 Movie). In contrast, the periods of the two areas were comparable in vehicle-treated cultures (Repeated measures two-way ANOVA, Rostral area vs Caudal area; F(1,10) = 8.2, p = 0.017, Interaction; F(1,10) = 7.5, p = 0.021. Post-hoc Bonferroni test; Rostral area vs Caudal area, Vehicle; p = 1.0, FK; p = 0.0016). Further, we divided the SCN bioluminescence images into grids (64 × 64 μm) for detailed analyses (Fig 3C). In the vehicle-treated cultures, the phase differences among the acrophases of the circadian rhythm in each grid were maintained from day 1 to day 5 (Fig 3C, upper panel). In contrast, in FK-treated cultures the phase difference among grids gradually increased (Fig 3C, lower panel). This difference is quantitatively shown in Fig 3D, which compares acrophase SD (standard deviation) within the SCN slices for each day in culture. The SDs of the 2nd–5th acrophases of the Caudal area were significantly larger than those of the Rostral area (Multivariate comparison, Caudal vs Rostral; F(1,117) = 14, p = 0.0043, Peak 1st–5th; F(1,117) = 72, p < 0.001, Interaction; F(4,117) = 8.7, p < 0.001. Post-hoc Tukey test; Caudal vs Rostral; 1st, p = 0.1823; 2nd, p < 0.001; 3rd, p < 0.001; 4th, p < 0.001; 5th, p < 0.001, Vehicle vs FK; 1st, p = 0.9160; 2nd, p = 0.0296; 3rd, p < 0.001; 4th, p < 0.001; 5th, p < 0.001). This finding suggested that FK administration caused desynchrony among circadian rhythms in the SCN. We divided the SCN into smaller grids (32×32 μm) for further detailed analyses and visualized circadian period as a map. In FK-treated slices, we found the caudal region showed periods shorter than 24 hours (Fig 3E, S2B Fig). This region showing shorter periods (designated as short period region; SPR) occupied the caudal tip of the SCN and continued to the medial narrow area. In contrast, the circadian periods of other areas in the SCN were longer than 24 hours (designated as long period region; LPR).

## Effect of rostro-caudal separation on circadian period

Knowing the localization of SPR and LPR, we investigated which region is dominant when they are synchronized. Horizontal SCN slices with a thickness of 150 was also divided into rostral and caudal fragments by scalpel (n = 7, Fig 4A). All fragments were placed on one culture insert and bioluminescence was recorded by EMCCD cameras. We set ROIs on the edges of the caudal, rostral and intact SCN, designating them Rostral, Caudal, and Intact, respectively, and the circadian periods of the bioluminescence from each ROI was measured. The circadian periods of Caudal were significantly shorter than those of Rostral (Fig 4B–4D) and Intact (Repeated measures one-way ANOVA, Rostral vs Caudal; F(2,12) = 12, p = 0.0016, Post-hoc Bonferroni test; Intact vs Rostral, p = 1.0; Intact vs Caudal, p = 0.0042; Rostral vs Caudal, p = 0.0036). Simultaneously, we prepared SCN slices without dividing by scalpel, and

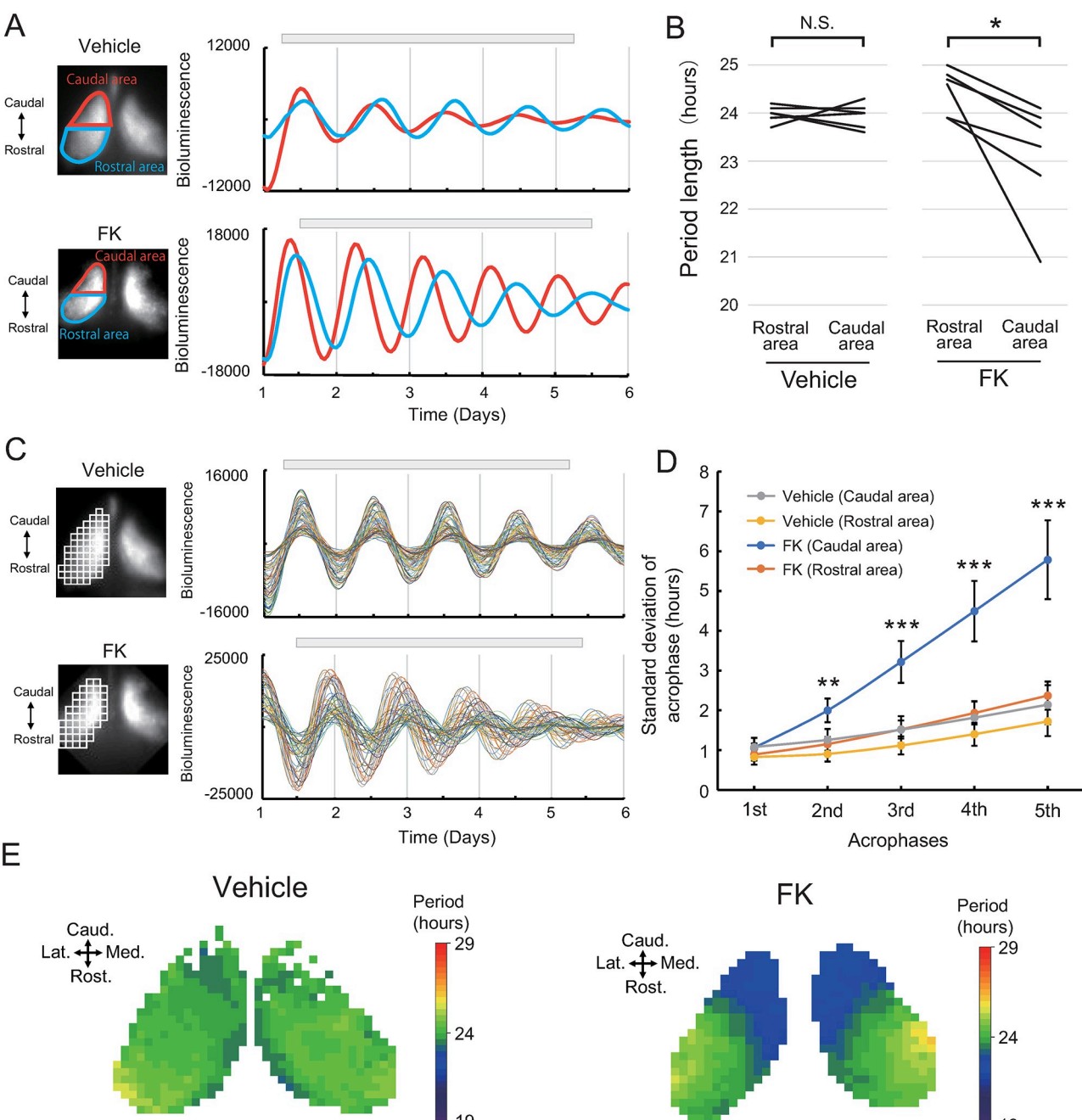

**Fig 3. Effect of forskolin on *Per2*::dLuc bioluminescence rhythm.** (A) Representative bioluminescence image of the SCN (left) and bioluminescence rhythm (right) of vehicle (upper panel) or FK-treated slices (lower panel). The average bioluminescence intensities of rostral (blue) and caudal (red) areas are plotted against time. Gray bars show the periods for curve fitting. The difference in period length between rostral and caudal areas increased in FK-treated slices. (B) Period lengths of the circadian rhythm in the rostral and caudal areas in the vehicle (n = 6) and FK-treated (n = 6) groups. The mean values of the bioluminescence periods of the rostral and caudal areas of the vehicle group were 24.0 ± 0.06 hours, 23.9 ± 0.1 hours, and in the FK-treated group, 24.1 ± 0.3 hours and 22.7 ± 0.5 hours, respectively (Mean ± SE). Repeated measures two-way ANOVA with post hoc Bonferroni test showed no significant difference between rostral and caudal areas of Vehicle group, but showed a significant difference between rostral and caudal areas of FK-treated group (*: $p < 0.05$, N.S.: not significant). (C) Small grids were used to divide the unilateral SCN, and bioluminescence data from all grids were plotted against time. With the vehicle, the period lengths were almost identical. In FK-treated slices, the circadian rhythms showed desynchrony compared with vehicle-treated slices. Gray bar: time used for curve fitting. Grid size: 64 μm. (D) The standard deviation of acrophase in each (1st–5th) cycle within a single SCN slice (mean ± SE). Acrophase was calculated by the fitted curve obtained from the bioluminescence of each grid. According to multivariate comparison and post-hoc Tukey test, there were significant differences between the rostral area and caudal area in phase variation of the 2nd–5th cycles (**: $p < 0.01$, ***: $p < 0.001$). (E) The period lengths of the circadian bioluminescence rhythms from grids were calculated by curve fitting. The period lengths of the vehicle-treated slice were similar, whereas the period lengths of the FK-treated slices showed shorter circadian rhythms in the caudal to medial grids than those in other grids. Med: medial, Lat: lateral. Grid size: 32 μm.

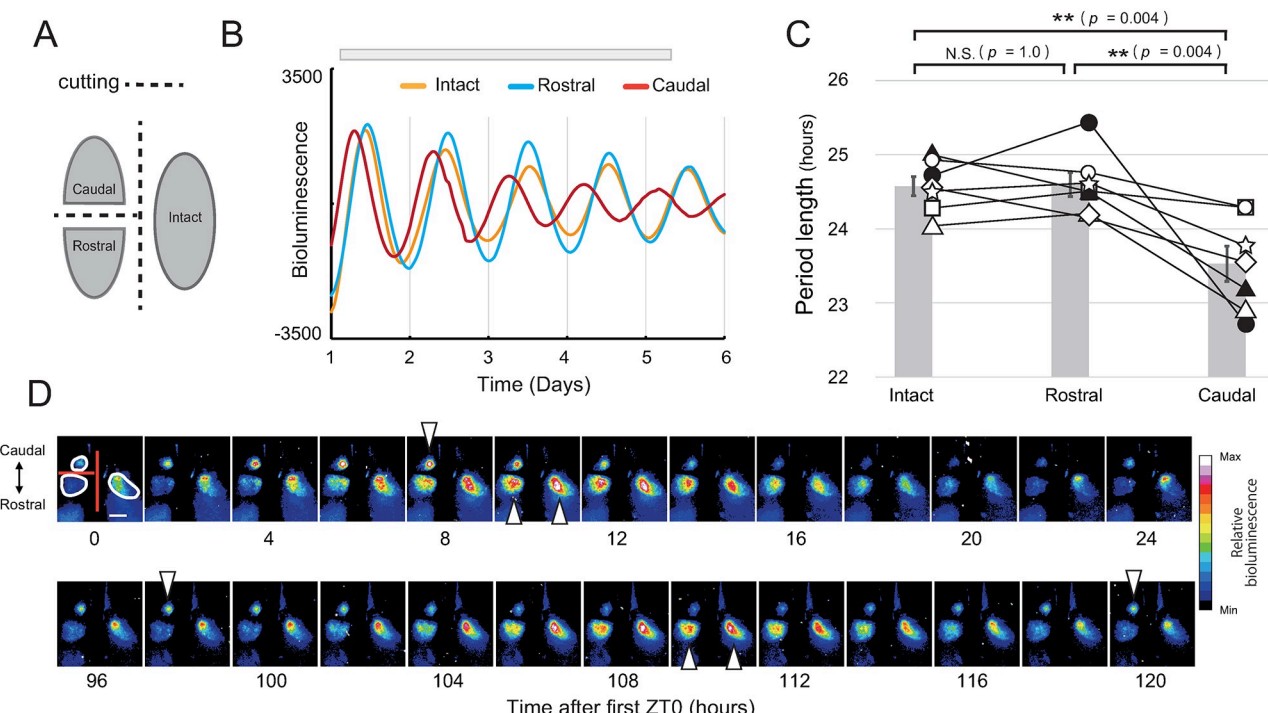

**Fig 4. Effect of rostro-caudal separation on *Per2*::dLuc bioluminescence rhythm.** (A) Schematic diagram of the dissected SCN. ROIs were set by outlines of Caudal, Rostral and Intact, and the circadian period of the bioluminescence from each ROI was measured. (B) One representative *Per2*::dLuc bioluminescence rhythm from three ROIs of a single slice. Gray bar indicates the period for curve fitting. (C) Circadian period of each ROI (n = 7). The period lengths were 23.5 ± 0.2, 24.6 ± 0.2, 24.6 ± 0.1, (Mean ± SE) in Caudal, Rostral and Intact, respectively. Repeated measures one-way ANOVA with post-hoc Bonferroni test: $^{**}p < 0.01$, N.S. = not significant. (D) Representative bioluminescence images of the SCN dissected by scalpel. White lines represent the outlines of Caudal, Rostral and Intact. White arrowheads indicate the peak phases of each fragment. Scale bar: 500 μm.

compared the circadian period with the intact side of the SCN dissected by scalpel (S3A Fig). We found that midline dissection of the SCN had no significant effect on the period length in horizontal SCN slices (S3B Fig).

## Discussion

In this study, by disrupting the synchrony among oscillating neurons, we found a region located mainly in the caudal area of the SCN showing circadian periods much shorter than 24 hours (SPR) in a horizontal slice culture of the SCN (Figs 2, 3 and S2 Fig). The direction of circadian phase wave propagation in the rat SCN detected by bioluminescence was from the SPR to other areas, which suggested that the SPR initiates the phase wave at the first step (Fig 1). Further, we dissected the SCN into rostral and caudal fragments by scalpel and found that the circadian period in the caudal fragment was much shorter than those in the rostral and intact SCN slices, which suggested that the circadian period of the rat SCN is determined by LPR rather than by SPR (Fig 4).

The localization of SPR in horizontal slices observed in the present study seems to be consistent with our previous study using coronal slice cultures [21]. In the previous study, we specified a narrow medial region of the SCN with a shorter *Per2* expression period (SPR) and found that the phase wave propagated from SPR to LPR [21]. In the present experiment, we also observed that the phase wave of *Per2* expression in SCN horizontal slices propagated from caudal to rostral and from medial to lateral (Fig 1C–1F), that is, from SPR to LPR. The narrow SPR at the middle of the SCN in the horizontal slice shown by FK treatment (Fig 3E) was

consistent with the morphological analysis of SPR in coronal sections in our previous study [21]. In addition, the direction of phase wave propagation was consistent between studies in that the wave started at the SPR and ended at the LPR. It is highly probable that the SPR observed in our previous study is identical to the SPR shown by the horizontal SCN slice analysis in the present study.

What mechanism binds the regions with distinct circadian periods? VIP and AVP are peptides that have been thoroughly investigated as substances synchronizing the oscillating neurons in the SCN [16, 35–37]. VIP is richly expressed among the neurons in the core of the SCN, and its receptor VPAC2 is mainly localized in the shell region of the SCN [15, 16]. VPAC2 gene deficient mice show desynchrony among oscillating neurons in the SCN [38]. In contrast, AVP-expressing neurons are mainly localized in the shell [25] and work to synchronize the oscillating neurons in the shell [26]. When dissected by scalpel, the caudal fragments contained the shell region and the rostral fragment contained both shell and core regions as shown in the in situ hybridization study showing the localization of VIP neurons as a core region marker (Fig 1B). AVP is abundantly expressed in the shell region of the SCN and contributes to maintaining the synchrony and phase difference among oscillating neurons in the SCN [15, 16]. Considering our present findings that there are regions with distinct intrinsic circadian periods within the SCN, synchronization of circadian oscillators by VIP and AVP contributes substantially to the circadian period of the SCN. On the other hand, Shinohara et al. [23] revealed that in rat suprachiasmatic nucleus slice cultures treated with antimitotic drugs that decreased the number of glial cells, the release of arginine vasopressin and vasoactive intestinal polypeptide showed different circadian periods. The finding suggests that the glial cells are also involved in the synchrony of oscillating neurons in the SCN.

Differences in the circadian periods between VIP- and AVP-neurons might be involved in the regional differences in circadian period. Noguchi et al. [25, 26] reported that AVP cells have intrinsically short circadian periods and are entrained by VIP cells. These studies suggest that AVP- and VIP-expressing neurons have distinct circadian periods [23, 25, 26, 36]. In the present study, we divided the horizontal SCN slice into rostral and caudal fragments. As shown by the in-situ hybridization study (Fig 1B), most of the *Vip*-expressing neurons were contained in the rostral fragment. Therefore, it is possible that the shortening of the circadian period of the caudal fragment is due to the removal of the VIP-expressing neurons. However, this hypothesis that VIP neurons and AVP neurons respectively generate long and short circadian rhythms seems to be inconsistent with our present and previous findings [21] that the shell region contains both SPR and LPR. AVP are rich in the shell region, so the entire shell region would generate short circadian rhythms. However, this contradiction may be explained by the uneven localization of AVP-expressing neurons in the shell. AVP-expressing neurons are dense in the medial region and sparse in the lateral region of the shell [39, 40]. If AVP-neurons generate shorter circadian periods, it is possible that partial region of the shell containing dense AVP-neurons generates short circadian periods compared with those in other shell regions.

The SPR has similar characteristics to the morning oscillator (MO) in that it is localized in the caudal region and in that the phase wave propagation starts there. Many studies have suggested the existence of a distinct morning oscillator (MO) and evening oscillator (EO) within the SCN, as the activity rhythms of rodents are separated into two components under certain LD conditions [41–43]. Jagota et al. [44] measured electrophysiological activity in horizontal SCN slices of hamsters under varying LD cycles and found two distinct peaks, possibly representing the MO and EO. In more recent studies, a bioluminescence reporter has been used to investigate the localization of EO and MO in the SCN. Inagaki et al. [28] showed two groups of oscillators coupled to the onset and end of activity (indicating EO and MO) in the mouse

SCN. Another study by Yoshikawa et al. [45] mapped the localization of the two oscillators on the horizontal SCN slices in which the MO is located in the caudal tip of the SCN. In the present study, we found a region with a short circadian period in the caudal region of the SCN, and this region seemed to initiate the phase wave propagation. The finding is consistent with the properties of the MO reported previously [45]. Therefore, the present study suggested that SPR observed in the present study might be identical with the MO shown in other studies [43, 45].

In the present study, the vehicle group also showed differences in the circadian periods within the SCN (S2B Fig). The region that showed the short circadian period was mainly located in the caudal region where the SPR is, indicating that there was desynchrony even without FK. It is possible that the fragments of the SCN lost by preparing horizontal slices might be also necessary for the synchrony of oscillating neurons in the horizontal slices of the SCN. As shown in Fig 1A, the SCN slice culture did not contain the entire SCN. The lost fragment of SCN was essential for keeping the entire SCN synchronized. In addition, the retinohypothalamic tract, a neural projection from the retina to SCN, was also lost in this slice culture. It is possible that this loss also contributed to the desynchrony with vehicle treatment. Such structural disruptions and loss of components might be a limitation of slice culture experiments.

In conclusion, we analyzed the regional circadian period difference in the rat SCN and found that the phase wave propagates from the SPR to LPR. Further, we found that the circadian period of the caudal region is entrained by the rostral region of the SCN, which constitutes the overall integrated period of the whole SCN. The localization of the SPR and the direction of the phase wave propagation suggested that the SPR in the caudal region of the SCN may be identical to the MO of the two-oscillator model. The relationships between the SPR/LPR and MO/EO should be investigated further.

## Supporting information

**S1 Fig. *Per2*::dLuc bioluminescence rhythm by PMT from consecutive cultured coronal SCN slices.** A representative of the six specimens is shown in Fig 2A. Data from five other SCNs are shown. The gray lines and black dotted lines indicate the detrended wave forms and fitted curves, respectively.
(TIF)

**S2 Fig. Effect of FK on *Per2*::dLuc bioluminescence rhythm.** (A) Time series of bioluminescence images from SCN horizontal slices treated with vehicle (DMSO, upper panel) and forskolin (FK, lower panel). The numbers below the pictures indicate projected ZT (ZT, zeitgeber time). Scale bar: 500 μm. (B) Grid analysis of circadian periods of *Per2*::dLuc bioluminescence rhythms. Grid size: 32 μm.
(TIF)

**S3 Fig. Effect of separating right and left SCN in a horizontal slice.** (A) ROIs were set on the SCN with and without dissection. Mean bioluminescence rhythm from each ROIs were measured. The periods of bioluminescence rhythms from the SCN without dissection (left (a) and right (b), left picture, Dissection(-), n = 5) and from unilateral SCN with dissection along the midline but without rostro-caudal dissection ((c), right picture, Dissection(+), n = 7). Dashed lines in the right picture indicate the dissection lines. (B) Statistical analysis between Dissection(-) and Dissection(+) groups. Bioluminescence period was 24.8 ± 0.14, 24.9 ± 0.16 hours for Dissection(-) group (Mean ± SE, Left and Right respectively), and 24.6 ± 0.13 hours for Dissection(+) group (Intact). No significant difference was found by repeated measures one-way ANOVA with a post hoc Bonferroni test between right and left SCN without dissection (a

and b) and with dissection (c) (repeated measures one-way ANOVA; F(2,8) = 1.4, $p$ = 0.31: post-hoc Bonferroni test; a vs. b, $p$ = 1.0, a vs. c, $p$ = 1.0, b vs. c, $p$ = 0.42).
(TIF)

**S1 Movie. Bioluminescence imaging of a horizontal SCN slice without any treatment.** The number in the lower right corner indicates the elapsed time from the start of the measurement.
(AVI)

**S2 Movie. Bioluminescence imaging of a horizontal SCN slice with FK treatment.** The number in the lower right corner indicates the elapsed time from the start of the measurement.
(AVI)

## Acknowledgments

We thank Dr. Seiichi Hashimoto (Astellas Pharma Inc.) for the supply of *Per2-dLuc* transgenic rats. We also thank Ms. E. Uemukai and Ms. Y. Yoshikawa for the care and management of laboratory animals and laboratory materials.

## Author Contributions

**Conceptualization:** Tadamitsu Morimoto, Mamoru Nagano, Yasufumi Shigeyoshi.

**Data curation:** Tadamitsu Morimoto, Tomoko Yoshikawa, Mamoru Nagano.

**Formal analysis:** Tadamitsu Morimoto, Tomoko Yoshikawa, Yasufumi Shigeyoshi.

**Funding acquisition:** Tomoko Yoshikawa, Mamoru Nagano, Yasufumi Shigeyoshi.

**Supervision:** Yasufumi Shigeyoshi.

**Writing – original draft:** Tadamitsu Morimoto.

**Writing – review & editing:** Tadamitsu Morimoto, Tomoko Yoshikawa, Mamoru Nagano, Yasufumi Shigeyoshi.

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
