## [Decision Letter · Decision Letter 0]

30 Jun 2022

PONE-D-22-16293Regionality of short and long period oscillators in the suprachiasmatic nucleus and their manner of synchronizationPLOS ONE

Dear Dr. Shigeyoshi,

Thank you for submitting your manuscript to PLOS ONE. After careful consideration, we feel that it has merit but does not fully meet PLOS ONE’s publication criteria as it currently stands. Therefore, we invite you to submit a revised version of the manuscript that addresses the points raised during the review process.

We look forward to receiving your revised manuscript.

Kind regards,

Shin Yamazaki, Ph.D.

Section Editor

PLOS ONE

Journal Requirements:

Additional Editor Comments:

Your manuscript was reviewed by two expert reviewers. Although both reviewers recommended accepting your manuscript with minor revisions, both expressed several concerns. Please address all of the concerns and incorporate specific suggestions that both reviewers made. I also noticed that the sample size (the number of biological replications) of the experiments shown in Fig. 1 and 4 is not clearly stated. Please state the sample size explicitly. Also, please follow the ARRIVE 2.0 guidelines

(Experimental animals | ARRIVE Guidelines

The ARRIVE guidelines 2.0: Updated guidelines for reporting animal research | PLOS Biology)

and provide the necessary information (e.g., age and sex of rat used, type of anesthesia used, etc.) in your manuscript. PLoS ONE doesn’t perform language editing and the accepted manuscript will be published as-is. One of the reviewers indicated several places where grammar needs improvement. Please consider using an independent scientific editing service.

Reviewers' comments:

Reviewer's Responses to Questions

**Comments to the Author**

1. Is the manuscript technically sound, and do the data support the conclusions?

Reviewer #1: Yes

Reviewer #2: Yes

2. Has the statistical analysis been performed appropriately and rigorously? 

Reviewer #1: Yes

Reviewer #2: Yes

3. Have the authors made all data underlying the findings in their manuscript fully available?

Reviewer #1: Yes

Reviewer #2: No

4. Is the manuscript presented in an intelligible fashion and written in standard English?

Reviewer #1: Yes

Reviewer #2: Yes

5. Review Comments to the Author

Reviewer #1: In this study by Morimoto et al., the authors aimed to 1) localize regions in the SCN with different circadian periods and, 2) determine which, if any, region “dictates” the overall period of the whole SCN. To do this, the authors used population and single-cell bioluminescence imaging of horizontal SCN slices from transgenic rats expressing a bioluminescent reporter of the core clock gene Per2. They found a phase wave of bioluminescence that originated in the caudal SCN, which predominantly expressed Vip mRNA and was phase-advanced compared to the rostral SCN. In coronal SCN slices, the caudal SCN also exhibits shorter periods of bioluminescence than the rostral SCN. However, in horizontal SCN slices, the authors find no difference in period between the rostral and caudal SCN (treated with vehicle). They do observe that the caudal SCN in the horizontal slice exhibits a shorter period than the rostral SCN when the slice is desynchronized with forskolin. Finally, they find that when the horizontal SCN is physically separated, the caudal SCN has a much shorter period than the rostral SCN. Surprisingly, the period of the intact SCN is comparable to that of the rostral, not caudal SCN. The authors claim that this suggests the rostral “long-period region” of the SCN determines the overall period of the SCN. In general, the manuscript is well written. There are a few places in the manuscript where grammar needs improvement to help the reader better understand the text, but overall the writing is acceptable. The statistics also seem acceptable. This manuscript will be of interest to circadian biologists and, perhaps, neuroscientists in general. The manuscript will be improved when the authors address the following concerns.

Fig 1C The authors should quantify the propagation of the phase wave in the horizontal slice along the rostral-caudal, medial-lateral axes. One possible way to do this is to plot a center-of-luminescence trajectory as in Brancaccio et al. Neuron 2013, Patton et al. Nat Comm 2020, and others.

Figs 3A, B, E The authors use forskolin to disrupt intracellular synchrony in a horizontal SCN slice and observe that the caudal SCN has a shorter circadian period than the rostral SCN. However, in vehicle treated horizontal slices, the period in the rostral SCN is not significantly different from the caudal SCN. How does this fit with the conclusion from Fig. 2 that in coronal slices, the caudal SCN has a shorter circadian period than the rostral SCN? Are coronal slices “more desynchronized,” similar to the effect of adding forskolin to a horizontal slice? The phase maps shown in Supplementary Fig 2B suggest that in some horizontal slices treated (c, d, perhaps e), the caudal SCN has a shorter period than the rostral SCN, but this is not quantified.

Figs 3C,D Was acrophase synchronicity different between rostral and caudal ROIs? The authors clearly show that there is more phase dispersal in forskolin treated slices, but this is perhaps unsurprising given the authors’ previously published work (Koinuma et al. EJN 2013, Sujino et al. Sci Rep 2018). It would be exciting to see if the observed rostral-caudal period difference in the horizontal slice is accompanied by a rostral-caudal difference in phase dispersal.

Reviewer #2: This report provides valuable insight into the organization of circadian rhythms generated by cells within the rat SCN. The focus is on two major structures, the core and shell, which differ in their major cell constituents, and additional subregions in horizontal brain slice cultures. The study used VIP expression to delimit the core, and AVP neurons are enriched in the shell. Nevertheless, the SCN contains several other neuronal types and glial cells in both regions, which were not addressed in this study, and these cells are also capable of generating circadian rhythms. More details are needed on the choice of procedures used and the possible implications of the results to better understanding the multiple timing functions of the SCN.

Major questions:

The point raised by the authors that the caudal SCN region with a shorter intrinsic circadian period may coordinate the timing of circadian rhythms throughout the SCN deserves additional discussion. For example, what are the implications for entrainment, and are there studies in mice or other mammals supporting this rhythm pattern?

An additional point needing discussion is on the relevance of these in vitro results to the SCN of the intact animal and its integration with the rest of the brain. Are the authors certain that the differences in period and phase observed between SCN regions is not an artifact of the isolation of neural tissue, separating it from other SCN regions and SCN afferents that could serve in coordinating rhythms in SCN subregions? Is there any evidence that the SCN in the animal also shows these patterns? Do any ablation studies in rats or hamsters support the connectivity of rhythmic regions and phase wave propagation described here?

Line 121, What was the dorsal-ventral position for the horizontal sections? How much of the retinohypothalamic tract remained, or was the section above this tissue?

Line 159, Excluding grids from the analysis where rhythm data deviated from the fitted cosine wave seems to conflict with providing an accurate characterization of the SCN’s behavior. Why was this assumption that the circadian rhythm must match a cosine wave used here? Circadian rhythms have many different waveforms. It is important to justify why this step was done and indicate how much data was excluded. Are the authors certain that they are not missing important circadian properties? Imagine a cell type or group of cells that are only active briefly during the day but provide an important role. They would not easily match the fitted function.

Line 238, Why was forskolin used to disrupt circadian rhythms in this study and the previous one? Why was it preferred instead of an agent that acts more specifically like known neurotransmitters or peptides used in intercellular communication between SCN cells?

Minor concerns:

Line 138, Provide EMCCD camera gain and exposure times used.

Line 269, This phrase seems vague and needs more clarity and description: “…showed different circadian periods by anti-mitotic treatment…”

The study appears to be focused on certain SCN neurons. What might be the role of glial cells or other neuron types in these coordinated circadian rhythms? Could they have generated some of the bioluminescence signal measured, and how could that affect the seeming interpretation that VIP an AVP cells are most responsible for the observed phenomena?

6. PLOS authors have the option to publish the peer review history of their article (what does this mean?). If published, this will include your full peer review and any attached files.

Reviewer #1: No

Reviewer #2: No

---

## [Author Response · Author response to Decision Letter 0]

31 Aug 2022

Author‘s Response to Reviewers

First of all, we appreciate reviewers for their constructive and meaningful suggestions. According to reviewers’ suggestion, we revised --manuscript and Fig 1D,1E and Fig 3D. Point to point responses are shown as follows. Comments by editor and reviewers are attached with ## in front of the paragraph and responses by authors are without any signs.

Response to Editor:

## I also noticed that the sample size (the number of biological replications) of the experiments shown in Fig. 1 and 4 is not clearly stated. Please state the sample size explicitly. Also, please follow the ARRIVE 2.0 guidelines

(Experimental animals | ARRIVE Guidelines

The ARRIVE guidelines 2.0: Updated guidelines for reporting animal research | PLOS Biology)

and provide the necessary information (e.g., age and sex of rat used, type of anesthesia used, etc.) in your manuscript.

Thank you very much for pointing out. We have corrected the manuscript. (Line 89, 178, 310)

Reviewer #1

## Fig 1C The authors should quantify the propagation of the phase wave in the horizontal slice along the rostral-caudal, medial-lateral axes. 

Thank you very much for pointing out. Fig 1E, which we originally presented (upper panel), was dividing the horizontal SCN into three sections in the rostral-caudal directions, and was indicating by the phase shift of each bioluminescence that the phase wave was directed from the caudal to the rostral direction. Following reviewer’s suggestion, we have added a figure (Fig 1E, lower panel) that clarifies the progression of the phase wave in the medial-lateral direction. These figures clearly show the propagation of the phase wave from the medial to the lateral direction. In addition, we have added a comment to the manuscript that Fig 1F clearly shows that the phase wave propagates from the green region to the brown region regardless of how the region is divided.

Added as follows: (Line 211-216) The phase in the caudal ROI (red) are advanced compared to those of the middle (yellow) and rostral (blue) fragments (upper panel). The phases in the medial ROIs (red) are advanced compared to those of the middle (yellow) and lateral (blue) fragments (lower panel). (F) Phase map of 1st acrophase (after the first pZT0) of the SCN. This figure indicates that the phase wave propagates from the green region to the brown region.

## Figs 3A, B, E The authors use forskolin to disrupt intracellular synchrony in a horizontal SCN slice and observe that the caudal SCN has a shorter circadian period than the rostral SCN. However, in vehicle treated horizontal slices, the period in the rostral SCN is not significantly different from the caudal SCN. How does this fit with the conclusion from Fig. 2 that in coronal slices, the caudal SCN has a shorter circadian period than the rostral SCN? Are coronal slices “more desynchronized,” similar to the effect of adding forskolin to a horizontal slice? 

Thank you for the valuable comment. In the study using coronal slices, rostrocaudal connection was physically disrupted in coronal slices. In such conditions, we found that the circadian period observed in the caudal SCN slice was shorter than in others (Fig. 2B). Therefore, it is highly probable that, in the horizontal SCN slices which showed synchrony among oscillating neurons, the rostrocaudal connection was preserved. Based on these findings, we have speculated that the communication between neurons in the horizontal plane plays an important role in maintaining synchrony within the SCN. We consider that, in the coronal slices, the rostral-caudal communication has been interrupted, revealing the endogenous intrinsic circadian periods in the caudal SCN.

## The phase maps shown in Supplementary Fig 2B suggest that in some horizontal slices treated (c, d, perhaps e), the caudal SCN has a shorter period than the rostral SCN, but this is not quantified.

Thank you for pointing this out. The observation of the regions showing short periods in the vehicle-treated horizontal SCN slices suggests that some caudal small regions of the SCN slices are out of synchrony with other regions. We speculate that the reason for this is that the slices do not contain the entire SCN as shown in Figure 1A; we believe that the loss of a portion of the SCN makes it difficult to achieve synchronization of the entire SCN contained in the slices. In vehicle-treated horizontal SCN slices, the desynchronized subregions vary widely from section to section, so we have not quantified them beyond the period mapping of each section to show the period. This is one of the limitations of ex vivo studies. We have added these contents above in the Discussion section.

Added as follows: (Line 414-419) In the present study, the vehicle group also showed differences in the circadian periods within the SCN (Supplementary Figure S2B). The region that showed the short circadian period was mainly located in the caudal region where the SPR is, indicating that there was desynchrony even without FK. It is possible that the fragments of the SCN lost by preparing horizontal slices might be also necessary for the synchrony of oscillating neurons in the horizontal slices of the SCN. 

## Figs 3C,D Was acrophase synchronicity different between rostral and caudal ROIs? The authors clearly show that there is more phase dispersal in forskolin treated slices, but this is perhaps unsurprising given the authors’ previously published work (Koinuma et al. EJN 2013, Sujino et al. Sci Rep 2018). It would be exciting to see if the observed rostral-caudal period difference in the horizontal slice is accompanied by a rostral-caudal difference in phase dispersal.

Thank you for the valuable comment. According to the reviewer’s comment, we re-analyze the phase synchrony of rostral and caudal ROIs separately and found significant difference. As shown in revised Fig. 3D. it clearly showed that the short period region of the caudal side in the FK-treated slices is accompanied by a large SD of the phase compared to rostral side. 

Reviewer #2

Major questions:

## The point raised by the authors that the caudal SCN region with a shorter intrinsic circadian period may coordinate the timing of circadian rhythms throughout the SCN deserves additional discussion. For example, what are the implications for entrainment, and are there studies in mice or other mammals supporting this rhythm pattern?

Thank you very much for the comment. We have data in the SCN of Per2::LUC knock-in mice, which indicates regional period differences within the SCN. However, the data is preliminary so far. We consider the analysis of other species an issue for further investigation.

## An additional point needing discussion is on the relevance of these in vitro results to the SCN of the intact animal and its integration with the rest of the brain. Are the authors certain that the differences in period and phase observed between SCN regions is not an artifact of the isolation of neural tissue, separating it from other SCN regions and SCN afferents that could serve in coordinating rhythms in SCN subregions?

Thank you very much for pointing this out. In SCN slice cultures, not all of the components of the SCN are included in one section. The input and output signals to the SCN, including the retinohypothalamic tract, are cut off. Therefore, we cannot rule out the possibility that the circadian period inside the SCN is influenced by the SCN cutting. We recognize this as a limitation of the present experiment. However, in our experiment, we mainly focused on the endogenous circadian period of neurons and the synchronization among neurons within the SCN. Therefore, we consider that the blockade of the synchronization inside the SCN not only by FK administration, but also by the physical blockade of intercellular signal transduction by slicing or knife-cut, have revealed the endogenous circadian rhythm and signaling pathway within the SCN. We have added these contents above in the Discussion section.

Added as follows: (Line 414-424) In the present study, the vehicle group also showed differences in the circadian periods within the SCN (Supplementary Figure S2B). The region that showed the short circadian period was mainly located in the caudal region where the SPR is, indicating that there was desynchrony even without FK. It is possible that the fragments of the SCN lost by preparing horizontal slices might be also necessary for the synchrony of oscillating neurons in the horizontal slices of the SCN. As shown in Figure 1A, the SCN slice culture did not contain the entire SCN. The lost fragment of SCN was essential for keeping the entire SCN synchronized. In addition, the retinohypothalamic tract, a neural projection from the retina to SCN, was also lost in this slice culture. It is possible that this loss also contributed to the desynchrony with vehicle treatment. Such structural disruptions and loss of components might be a limitation of slice culture experiments.

## Is there any evidence that the SCN in the animal also shows these patterns? Do any ablation studies in rats or hamsters support the connectivity of rhythmic regions and phase wave propagation described here?

Thank you very much for pointing this out. The phase wave propagation within the SCN has been extensively studied especially in Mouse. Mathematical models have also been used to analyze the results. On the other hand, we are not familiar with the results of studies on hamsters. We consider that the ablation studies of other species are an issue for further investigation.

## Line 121, What was the dorsal-ventral position for the horizontal sections? How much of the retinohypothalamic tract remained, or was the section above this tissue?

Thank you very much for the comment. When we prepared horizontal slices, we cut from the ventral side of the SCN and always cut sections in a thickness of 150 µm from where the optic nerve and optic chiasm just barely remained. Based on this, it is probable that the horizontal SCN slices are separated from the RHT. We added these contents in Discussion session.

Added as follows: (Line 414-424) In the present study, the vehicle group also showed differences in the circadian periods within the SCN (Supplementary Figure S2B). The region that showed the short circadian period was mainly located in the caudal region where the SPR is, indicating that there was desynchrony even without FK. It is possible that the fragments of the SCN lost by preparing horizontal slices might be also necessary for the synchrony of oscillating neurons in the horizontal slices of the SCN. As shown in Figure 1A, the SCN slice culture did not contain the entire SCN. The lost fragment of SCN was essential for keeping the entire SCN synchronized. In addition, the retinohypothalamic tract, a neural projection from the retina to SCN, was also lost in this slice culture. It is possible that this loss also contributed to the desynchrony with vehicle treatment. Such structural disruptions and loss of components might be a limitation of slice culture experiments.

## Line 159, Excluding grids from the analysis where rhythm data deviated from the fitted cosine wave seems to conflict with providing an accurate characterization of the SCN’s behavior. Why was this assumption that the circadian rhythm must match a cosine wave used here? Circadian rhythms have many different waveforms. It is important to justify why this step was done and indicate how much data was excluded. Are the authors certain that they are not missing important circadian properties? Imagine a cell type or group of cells that are only active briefly during the day but provide an important role. They would not easily match the fitted function.

Thank you for the valuable comment. We agree with the point that there could be cell types with non-cosine type circadian rhythms. In this study, however, we only focused on the cells that fit the cosine curve, since the percentage of ROIs excluded from the analysis was less than 1% for both Vehicle-treated and FK-treated groups. The low percentage of exclusion may be attributed to the size of the ROIs (32 µm × 32 µm) which is much bigger than a cell size, and the relatively moderate condition (r > 0.6). In addition, because we analyzed the bioluminescence intensities with grids larger than a single cell in size, it is possible that we were not able to recognize those cells which did not fit the cosine curve. Therefore, in this study, we only focused on the cells that fit the cosine curve and used the curve fitting including cosine within the equation. In the future, we believe that further improvement in spatial and temporal resolution may allow us to identify special groups of cells that do not fit the cosine wave.

## Line 238, Why was forskolin used to disrupt circadian rhythms in this study and the previous one? Why was it preferred instead of an agent that acts more specifically like known neurotransmitters or peptides used in intercellular communication between SCN cells?

We thank you for the comments. Attempts were made to obtain desynchrony in the rat SCN using MDLs (inactivators of adenylate cyclase) and TTX in the SCN, but desynchronization could not be achieved in the rat SCN even with MDL and TTX. It is because we could not determine the appropriate concentration to detect desynchronization. MDL and TTX made the circadian oscillations of each oscillating neuron of the rat SCN damp with low concentration compared with mouse SCN. Therefore, we used FK, which has been employed in previous reports, to attain desynchronization in the SCN. In a previous paper, we showed that FK administration does not affect the endogenous circadian rhythm of the cell line (Koinuma et al. European J Neurosci 2013 ). Further, in this paper, we also observed that there is a difference in the circadian periods in rostral and caudal coronal slices as shown in Fig.2B. We speculate that FK disrupts intercellular synchrony in the SCN, possibly via a sustained increase in cAMP concentration.

Minor concerns:

## Line 138, Provide EMCCD camera gain and exposure times used.

Thank you for pointing out. The cameras used and their gains are listed below. We have also added to the manuscript.

Multiversa with an EMCCD camera (iXon 897), Exposure：59 min., Em gain value : 500, Binning : 1×1

LUMINOVIEW with an EMCCD camera (C9100-23B), Exposure : 29～59min., Sensitivity gain : 150~200 (exposure 29min.), 100~150 (exposure 59min.), Gain : 1

Cellgraph with an EMCCD camera (AB-3000), Exposure : 59min., Electron Multiplier Gain : 300, Pre-Amplifier Gain : 1.0

## Line 269, This phrase seems vague and needs more clarity and description: “…showed different circadian periods by anti-mitotic treatment…”

Thank you for pointing out. Our quotation was not accurate. We have corrected as follows. (Line: 374-378)

(Before revision)

“Shinohara et al.[23] reported that AVP and VIP release from the cultured SCN showed different circadian periods by anti-mitotic treatment, and suggested that both AVP and VIP release from the SCN are under different circadian oscillators bearing different circadian periods.”

(After revision)

“Shinohara et al.[23] revealed that the rat suprachiasmatic nucleus slice culture with a treatment of antimitotic drugs that suppress the mitosis of glial cells, circadian rhythms in the release of arginine vasopressin and vasoactive intestinal polypeptide showed different circadian periods. The finding suggests that the glial cells are also involved in the synchrony of oscillating neurons in the SCN.”

## The study appears to be focused on certain SCN neurons. What might be the role of glial cells or other neuron types in these coordinated circadian rhythms? Could they have generated some of the bioluminescence signal measured, and how could that affect the seeming interpretation that VIP and AVP cells are most responsible for the observed phenomena?

It is highly probable that glial cells in the SCN have some functions as a part of the circadian oscillator. In the revised manuscript, we cited Shinohara’s study that shows the importance of glia for the synchronization. We added the sentences below.

(Line: 374-378) “Shinohara et al.[23] revealed that the rat suprachiasmatic nucleus slice culture with a treatment of antimitotic drugs that suppress the mitosis of glial cells, circadian rhythms in the release of arginine vasopressin and vasoactive intestinal polypeptide showed different circadian periods. The finding suggests that the glial cells are also involved in the synchrony of oscillating neurons in the SCN.”

Also, it is possible many factors are thought to be involved in the synchronization of the SCN other than the AVP and VIP neurons but also the GRP neuron and neurons expressing various neurotransmitters and neuromodulators. We consider that the analysis of the contributions of glial cells and other neurons to the circadian rhythm is an issue for further investigation.

We sincerely appreciate your kind but critical readings and meaningful suggestions.

---

## [Decision Letter · Decision Letter 1]

19 Sep 2022

PONE-D-22-16293R1Regionality of short and long period oscillators in the suprachiasmatic nucleus and their manner of synchronizationPLOS ONE

Dear Dr. Shigeyoshi,

Your revised manuscript was reviewed by the same reviewers who reviewed your original manuscript.

Although both reviewers recommended accepting your current manuscript for publication in PLoS ONE, reviewer #2 indicated two concerns. The reviewer suggested changing the statement in the Abstract. If you agree with this suggestion, please make the change accordingly. Also, please provide a specific plan for how you will make the original data available. 

We look forward to receiving your revised manuscript.

Kind regards,

Shin Yamazaki, Ph.D.

Section Editor

PLOS ONE

Journal Requirements:

Reviewers' comments:

Reviewer's Responses to Questions

**Comments to the Author**

1. If the authors have adequately addressed your comments raised in a previous round of review and you feel that this manuscript is now acceptable for publication, you may indicate that here to bypass the “Comments to the Author” section, enter your conflict of interest statement in the “Confidential to Editor” section, and submit your "Accept" recommendation.

Reviewer #1: All comments have been addressed

Reviewer #2: All comments have been addressed

2. Is the manuscript technically sound, and do the data support the conclusions?

Reviewer #1: Yes

Reviewer #2: Partly

3. Has the statistical analysis been performed appropriately and rigorously? 

Reviewer #1: Yes

Reviewer #2: Yes

4. Have the authors made all data underlying the findings in their manuscript fully available?

Reviewer #1: Yes

Reviewer #2: No

5. Is the manuscript presented in an intelligible fashion and written in standard English?

Reviewer #1: Yes

Reviewer #2: Yes

6. Review Comments to the Author

Reviewer #1: (No Response)

Reviewer #2: Relative to my response to Review Question 2: To allow for other possible interpretations of the data, I suggest that the authors qualify or moderate the statement in the Abstract "We found that the longer circadian rhythm of the rostral region entrains the circadian rhythm in the caudal region". A better statement might read "... the rostral region appears to entrain the circadian rhythm in the caudal region". I am concerned that a rigorous test of entrainment was not performed, which would require more sophisticated control of the interactions between the oscillators including a repeated control and release from entrainment and assessment of subsequent phase. Also, I did not see a statement saying that all the data are available or would be made available in a data repository if the paper is accepted for publication.

7. PLOS authors have the option to publish the peer review history of their article (what does this mean?). If published, this will include your full peer review and any attached files.

Reviewer #1: No

Reviewer #2: No

---

## [Author Response · Author response to Decision Letter 1]

4 Oct 2022

Author's Response to Reviewers

First of all, we appreciate reviewers for their constructive and meaningful suggestions. According to reviewers’ suggestion, we revised Abstract and added Data availability statement. Point to point responses are shown as follows. Comments by editor and reviewers are attached with ## in front of the paragraph and responses by authors are without any signs.

Reviewer #2

## Relative to my response to Review Question 2: To allow for other possible interpretations of the data, I suggest that the authors qualify or moderate the statement in the Abstract "We found that the longer circadian rhythm of the rostral region entrains the circadian rhythm in the caudal region". A better statement might read "... the rostral region appears to entrain the circadian rhythm in the caudal region". I am concerned that a rigorous test of entrainment was not performed, which would require more sophisticated control of the interactions between the oscillators including a repeated control and release from entrainment and assessment of subsequent phase.

Thank you for the valuable comment. We agree with your opinion and revised Abstract as follows: (Line 35-36) We found that the longer circadian rhythm of the rostral region appears to entrain the circadian rhythm in the caudal region.

## Also, I did not see a statement saying that all the data are available or would be made available in a data repository if the paper is accepted for publication.

Thank you very much for your comment. We added Data availability statement in the manuscript.

Added as follows: (Line 438 - 442) Data used to generate the figures and supplementary figures in this manuscript are accessible on a public figshare repository (10.6084/m9.figshare.21261324). Other data and further information are also available from the corresponding author on reasonable request.

We sincerely appreciate your kind but critical readings and meaningful suggestions.

---

## [Editor Report · Decision Letter 2]

6 Oct 2022

Regionality of short and long period oscillators in the suprachiasmatic nucleus and their manner of synchronization

PONE-D-22-16293R2

Dear Dr. Shigeyoshi,

We’re pleased to inform you that your manuscript has been judged scientifically suitable for publication and will be formally accepted for publication once it meets all outstanding technical requirements.

Kind regards,

Shin Yamazaki, Ph.D.

Section Editor

PLOS ONE
---

## [Editor Report · Acceptance letter]

10 Oct 2022

PONE-D-22-16293R2 

Regionality of short and long period oscillators in the suprachiasmatic nucleus and their manner of synchronization 

Dear Dr. Shigeyoshi:

I'm pleased to inform you that your manuscript has been deemed suitable for publication in PLOS ONE. Congratulations! Your manuscript is now with our production department. 

Kind regards, 

on behalf of

Dr. Shin Yamazaki 

Section Editor

PLOS ONE